# Transgenic Maize Has Insignificant Effects on the Diversity of Arthropods: A 3-Year Study

**DOI:** 10.3390/plants11172254

**Published:** 2022-08-30

**Authors:** Zhentao Ren, Muzhi Yang, Haopeng He, Yanjie Ma, Yijun Zhou, Biao Liu, Kun Xue

**Affiliations:** 1Country Key Laboratory of Ecology and Environment in Minority Areas, Minzu University of China, National Ethnic Affairs Commission, Beijing 100081, China; 2College of Life and Environmental Sciences, Minzu University of China, Beijing 100081, China; 3Nanjing Institute of Environmental Sciences, Ministry of Ecology and Environment, Nanjing 210042, China

**Keywords:** genetically modified organisms, maize, Bt, corn, biodiversity, arthropods, biosafety

## Abstract

In order to provide more evidence for the evaluation of the ecological risks of transgenic maize, arthropod population dynamics and biodiversity in fields planted with two kinds of transgenic maize (DBN9868, expressing the *PAT* and *EPSPS* genes, and DBN9936, expressing the *Cry1Ab* and *EPSPS* gene) were investigated by direct observation and trapping for three years. The recorded arthropod species belonged to 19 orders and 87 families, including Aphidoidea, Chrysomelidae, Coccinellidae, Chrysopidae and Araneae. The species richness, Shannon–Wiener diversity index, Pielou evenness index, dominance index and community similarity index of arthropod communities in maize fields were statistically analyzed, and the results showed that (1) the biodiversity difference of arthropod communities between transgenic maize and non-transgenic maize was smaller than that between different conventional cultivars; (2) the differences between ground-dwelling arthropod communities were less obvious than those between plant-inhabiting arthropod communities; and (3) Lepidoptera, the target pests of Bt maize, were not the dominant population in maize fields, and the dominant arthropod population in maize fields varied greatly between years and months. Combining those results, we concluded that the transgenic maize DBN9868 and DBN9936 had no significant effect on the arthropod communities in the field.

## 1. Introduction

Genetically modified (GM) crops have been used commercially for more than 20 years. GM maize is one of the most adopted GM crops in the world, with 60.9 million hectares planted in 2019, which is 31% of the global planted area (https://www.isaaa.org/ (accessed on 1 May 2022)). In 2020, three biosafety certificates for GM maize were granted by the Ministry of Agriculture, PR China. Although GM crops have been commercially planted at a large scale in recent decades, their biosafety is still controversial in certain countries [1,2]. One concern about GM crops is the potential negative impact on the diversity and abundance of non-target organisms and, subsequently, on ecosystem functions [3,4,5]. In maize field ecosystems, non-target arthropods provide important ecological functions such as biological control, recycling of organic matter, and pollination [6,7]. Prior to biosafety certification being granted, local field trials are required to assess the potential adverse effects of GM maize on agroecosystems, including arthropod communities, relative to a non-GM control.

A growing number of studies have revealed the non-target impact of GM maize on arthropod biodiversity in the field. To date, most studies have shown that insect-resistant and herbicide-tolerant traits, either single events or stacked products, do not adversely affect biodiversity, the populations of natural enemies, or other ecologically important non-target arthropods [6,8,9,10,11,12,13,14,15,16]. In recent years, some long-term studies have also supported this point of view. For example, a farm-scale evaluation of the impact of Bt maize (*Cry1Ab*) on non-target arthropods through a three-year study was reported, indicating that Bt maize had no negative effect on hemipteran herbivores (leafhoppers and planthoppers) and hymenopteran parasitic wasps (mymarids) [17]. A three-year study showed that Bt maize (*Cry1Ac*) had no significant negative effects on selected non-target arthropod diversity in the field [18]. Another three-year study also showed that Bt maize (*Cry1Ab* and *Cry2Ab*) was compatible with non-target arthropods [19]. Most studies compared transgenic maize with its parental line, but non-transgenic controls of different varieties were not included in the comparison.

Evaluation of GM crops follows the principles of precaution, familiarity and substantial equivalence, which are concepts arising from the knowledge and experience gained over time [20]. They are based on a series of comparative assessment approaches that consider the GM crop in the context of appropriate non-GM control and the known variation of the crop [21]. The mean values of the key characteristics of the GM crops are assessed against the range of values of the reference varieties or from the literature in the context of known values common for the crop [22]. Those comparative assessments can reveal the potential risks of the cultivation of GM crops [23,24]. This study uses comparative assessment approaches to evaluate the effects of GM crops on arthropod communities in the field. If the mean value of the arthropod diversity of a GM crop is beyond the value range of the cultivated crops, the detected differences are then used to assess whether they could potentially alter arthropod communities.

In this study, the arthropod biodiversity in plots with four kinds of maize was investigated in Northeast China from 2015 to 2017. The maize cultivars were commercial maize XianYu335, non-transgenic maize DBN318, transgenic maize DBN9868 providing tolerance to herbicides, and transgenic maize DBN9936 providing tolerance to herbicides and resistance to lepidopteran pests. In order to evaluate the impact of transgenic maize on arthropod communities, species richness (*S*), Shannon–Wiener index (*H*), dominance index (*C*), Pielou evenness index (*J*) and community similarity index (*C′*) of plant-inhabiting and ground-dwelling arthropods were analyzed. We had the following objectives: (1) to compare the effects of different types of transgenic maize on arthropod communities in the field; (2) to compare the effects of maize on arthropod communities on the plant and on the land; (3) to determine the dynamics of arthropod communities in maize fields in different months and years. The results will provide a statistical baseline and theoretical basis for the ecological risk assessment of transgenic crops.

## 2. Results

### 2.1. Composition of Arthropod Communities

The arthropod species in maize plots were investigated throughout the growing season in 2015, 2016 and 2017. A total of 89 families from 19 orders were identified and recorded, including insects, spiders and other arthropods (Appendix A).

A simplified food web, including the main plant-inhabiting and ground-dwelling arthropod species in maize fields, is shown in Figure 1. During all stages of the maize, *Rhopalosiphum maidis* and *Monolepta hieroglyphica* were the main plant-inhabiting herbivores. Lepidoptera herbivores rarely occurred, mainly including the Asian corn Borer (*Ostrinia furnacalis*), the cotton bollworm (*Helicoverpa armigera*) and the oriental armyworm (*Mythimna separata*). Natural enemies included predators and parasitoids. Ladybeetles (*Harmonia axyridis* and *Propylaea japonica*), minute pirate bugs (*Orius sauteri*), lacewings (*Chrysopa pallens*) and spiders (*Misumenops tricupiuatus* and *Synema globosum*) were the most frequent predators on maize. Meanwhile, parasitoids were difficult to find. The detritivores included fruit flies and psocids. Ants are omnivorous insects that protect aphid populations and repel ladybeetles on maize.

The number of species of ground-dwelling arthropods collected in pitfall traps was less than that of plant-inhabiting arthropods (Figure 1). The cricket (*Teleogryllus emma*) was the most abundant ground-dwelling herbivorous species. The ground-dwelling predators included the wolf spider (*Pirata subpiraticus*) and ground beetles (*Chlaenius bioculatus* and *Dolichus halensis*). The main scavenging insects on the ground were springtails (Collembola).

The cumulative number (summed data from all sampling dates) of arthropods over the growing season was analyzed (Figure 2, Appendix A). The most abundant herbivore groups were the aphids (Aphidoidea) and leaf beetles (Chrysomelidae), and important predator groups were ladybeetles (Coccinellidae), lacewings (Chrysopidae) and spiders (Araneae). The abundance of these groups varied highly between different years. For example, in 2015, the number of leaf beetles was the highest, and that of aphids was the second. However, this was reversed in 2016 and 2017, and the leaf beetle population almost disappeared in 2017.

A *t*-test was conducted to compare the total number of main arthropod groups in the plots of XY and NT, NT and T1, NT and T2 (Figure 2). In 2015, Chrysomelidae abundance on NT was significantly higher than that on XY (*p* = 0.047). In 2016, the Aphidoidea abundance on NT was significantly higher than that on XY (*p* = 0.0001). In 2017, Lepidopteran abundance on NT was significantly higher than that on T2 (*p* = 0.005). Other groups showed no significant difference.

### 2.2. The biodiversity Indices of Arthropods

Four indices were used for arthropod biodiversity analysis. The species richness (*S*) directly represents the biodiversity in an agroecosystem. The Shannon–Wiener biodiversity index (*H*) reflects species richness and community structure information, which is sensitive to changes in the rare species of a sample [25]. The Pielou evenness index (*J*) describes the relative abundance or ratio of individuals in certain species to the total number of individuals, which is often used in combination with *H* [26]. The dominance index (*C*), based on the Simpson’s index, is sensitive to changes in species abundance in the sample and less sensitive to species richness [27,28].

The *S* presented similar dynamics each year, regardless of the type of maize (Figure 3). The dynamics of *S* of the arthropod community in the plots between XY and NT, NT and T1, NT and T2 were compared using a *t*-test. The *S* of plant-inhabiting arthropods on NT was significantly different from that on XY for each pair of comparisons (26 August 2015, *p* = 0.047; 23 June 2016, *p* = 0.016). On 1 September 2016, the *S* of plant-inhabiting arthropods on NT was significantly higher than that on T1 (*p* = 0.013). In 2017, on sample date 7 September (*p* = 0.002), the *S* of plant-inhabiting arthropods on NT was significantly higher than that on T2. In 2015, on the sample dates of 21 July (*p* = 0.016) and 16 September (*p* = 0.047), the *S* of plant-inhabiting arthropods on XY was significantly higher than that on NT. There were no significant differences between the NT and XY, NT and T1, NT and T2 plots in terms of the *S* of ground-dwelling arthropods at each sampling time.

In this study, *H* also had similar dynamics each year, regardless of the maize varieties. There were eight significant differences between XY and NT on different sampling dates and no significant differences between NT and transgenic varieties (Figure 4).

For the Pielou evenness index, there were eight significant differences and two extremely significant differences between XY and NT in Student’s *t*-test, while there were two significant differences between NT and T1 and NT and T2, respectively (Appendix A). The *C* presented similar dynamics each year, regardless of the maize varieties. For the dominance index, there were seven significant differences that only existed in comparisons between XY and NT with Student’s *t*-test (Appendix A).

### 2.3. The Community Similarity Index of Arthropods

The community similarity index (*C′*) appears to be the best approximation of a linear measure of community relationship [29]. Thus, if the same species were in two communities, the *C′* index would be 1.0; if there were no species in common between the two communities, the *C′* index would be 0.0. The range from no resemblance to complete identity is appropriately covered by the range from 0.0 to 1.0. The closer the *C′* index is to 1.00, the more similar the two communities are. The *C′* of arthropod communities between XY and NT, NT and T1, and NT and T2 from 2015 to 2017 was analyzed (Figure 5). On the plants, 67% of the *C′* values of “XY vs. NT” were lower than the *C′* values of “NT vs. T (including T1 and T2)”, and the difference between them was significant according to a chi-square test (*n* = 52, *χ*^2^ = 6.231, *p* = 0.013). On the ground, 65% of the *C′* values of “XY vs. NT” were lower than the *C′* values of “NT vs. T”, and the difference between them was not significant according to a chi-square test (*n* = 26, *χ*^2^ = 2.462, *p* = 0.117). In total, the community similarity of arthropod between transgenic maize and non-transgenic maize (NT vs. T1; NT vs. T2) was smaller than that between different conventional cultivars (XY vs. NT).

## 3. Discussion

There was no evidence to suggest that transgenic crops are toxic to non-target arthropods in the field [10,30,31,32,33]. Our study showed that the biodiversity difference of arthropod communities between transgenic maize and non-transgenic maize was smaller than that between different conventional cultivars.

### 3.1. The Effects of Different Maize Traits on Arthropods

We hypothesized that the differences between arthropod communities between the different maize lines were not caused by transgenes but by differences in agronomic and phenotypic characteristics. For example, there were obvious differences in the characteristics of the husks of the ear between XY and NT. The husks of XY are relatively tight, while the husks of NT are loose (the looseness means aphids can get into the husks to feed, while their natural enemies, ladybeetles, cannot), which means aphids are outside of the control of ladybeetles and outbreaks occur more frequently. Some studies reported that slightly different agronomic and phenotypic characteristics were detected in the transgenic maize hybrids compared to conventional maize hybrids, but none of these phenotypic differences were expected to contribute to biological or ecological changes that would result in increased pest potential or ecological risks [34]. Therefore, the few significant differences in arthropod biodiversity between GM and non-GM maize plots do not indicate that GM crops have an adverse effect on the ecosystem.

### 3.2. The Effects of GM Maize on Arthropods

If the Bt protein is toxic to non-targets, we would expect those effects in Bt maize. For herbicide-tolerant maize, no effects on non-targets are hypothesized. In this study, T1 (*PAT* and *EPSPS* genes) and T2 (*Cry1Ab* and *EPSPS* genes) did not lead to significant differences in non-target arthropod biodiversity. As a result, Bt effects are unlikely to be present. Most of the plant-inhabiting arthropods (lepidopterans, leaf beetles, ladybeetles, aphids, stink bugs, biting insects, mites and thrips) are able to ingest the insecticide Bt Cry toxins expressed in GM crops [35,36,37,38,39]. Some studies have shown the presence of Cry toxins in some ground-dwelling arthropods (ground beetles, spider mites and Diptera larvae) through direct consumption of the maize or its residues [40,41,42]. However, the available literature does not provide evidence to support the claim that unexpected interactions between different Bt proteins may lead to adverse effects on non-target species [43]. Our results also demonstrated that arthropod communities are not affected by maize expressing *Cry1Ab*, *EPSPS* and *PAT*. This conclusion is consistent with the results of other studies on the effect of Bt maize on arthropods [6,44,45,46,47,48,49,50]. In addition, this study showed that the effect of different maize types on ground-dwelling arthropods was less than that on plant-inhabiting arthropods. This may be because many ground-dwelling arthropods, such as springtails, crickets and ground beetles, depend more on the soil or a weedy environment than on maize plants.

Our results are consistent with other field studies that have reported that the abundance of arthropod species varies greatly between years and months [17,18], which might be a consequence of biotic and abiotic elements in complex agricultural ecosystems, including food uniformity, temperature variation, rainfall distribution and the presence of natural enemies [17,51]. In our study, there was no long-term stable food web in maize-based farmland ecosystems during the growing period. Compared with the significant effects of a complex climate, the effects of GM maize on arthropod communities in the field appear negligible.

### 3.3. The Effects of GM Maize on the Food Web

Agroecosystems are simple but vulnerable to environmental factors. Multitrophic interactions in communities and food webs are temporary [52]. Food web analyses can provide comprehensive information to understand the effect of GM crops on the structure of ecological communities [12]. Bt maize, expressing *Cry1Ab* insecticidal proteins, kills corn borer and other lepidopteran target pests. In theory, Bt maize can affect the food chain with lepidopterans. In this study, a very low density of lepidopterans was detected in all plots, less than 1% of herbivores. Some studies showed that the abundance of target lepidopteran European corn borer (*Ostrinia nubilalis*) reached a maximum of 0.2% of herbivores [12]. Thus, the effects of GM maize on the food chains and webs with lepidopterans should be assessed.

Aphids do not belong to the leaf-chewing defoliators because of their special feeding behaviors, sucking phloem sap from the leaves and buds. Compared with lepidopterans, Bt maize had little effect on the food chain of aphids and ladybirds. In this study, the differences in aphid populations between maize varieties (XY and NT) were greater than those between GM maize and its counterpart (T1/T2 and NT), which suggested that Bt maize had little effect on the aphid–ladybird food chain. This explains why Bt maize hardly affects arthropod communities in the field. At the same time, there is the question of whether it is necessary to plant Bt maize in areas with low lepidopteran pest densities.

## 4. Materials and Methods

### 4.1. Maize Materials

Four maize varieties, including non-transgenic cultivated variety XianYu335 (XY), non-transgenic recipient variety DBN318 (NT), transgenic maize DBN9868 (T1) and DBN9936 (T2), were provided by DNB Biotech Co. Ltd. (Beijing, China). XY is a commonly cultivated variety in Northeast China. NT is the non-transgenic recipient for both T1 and T2. T1, expressing the *PAT* (Phosphinothricin *N*-acetyltransferase) and *EPSPS* (5-enolpyruvylshikimate-3-phosphate synthase) genes, is a glyphosate- and glufosinate-tolerant variety. T2, expressing the *Cry1Ab* and *EPSPS* genes, is an insect-resistant and glyphosate-tolerant variety. Both DBN9868 (T1) and DBN9936 (T2) were declared safe via safety certificates, and our research played a role in that declaration. The biosafety certificate for DBN9936 (T2) was granted by the Ministry of Agriculture, PR China, on 21 January 2020. The expression levels of Cry1Ab, EPSPS and PAT proteins in maize materials, determined by ELISA, are shown in Appendix A, and the methods are described in the Appendix A.

### 4.2. Planting Plot Design

A set of four independent experiments was conducted on the test base in Yitong (Jilin Province, China) (125°26′18.22″ E; 43°14′8.94″ N) from June to September of each year, 2015 to 2017. No crops were planted in the field until the experiment started. XY, NT and T1 were planted in 2015 and 2016; XY, NT and T2 were planted in 2015 and 2017. Each variety was replicated three times on 150 m^2^ (10 m × 15 m), with corridors of 1.0 m between them (the plots design is shown in Appendix A). The experiment was carried out with a randomized blocks design. Maize was sown with a between-row spacing of 60 cm and a within-row spacing of 25 cm. No herbicides or pesticides were applied during the growth period of maize.

### 4.3. Investigation of Arthropods

The diagonal five-point sampling method was used to determine sampling sites. At each sampling site, 10 maize plants were investigated.

The arthropods on the plants were counted by direct observation. The investigations on arthropods were taken once a week from the 50th day after cultivation to harvest. The families, number and developmental stages of all the arthropods on plants were recorded. Some species that occur in large numbers were counted and collected for later identification in the laboratory. In addition, the food consumption behaviors of arthropods were observed in order to map the food web.

The ground-dwelling arthropods were collected with pitfall traps. Samples were taken once every 14 days from the 50th day after planting to harvest. There were five sample sites within each plot and three traps within each sample site. The traps were made of plastic cups (Ø7 cm × 8 cm) holding one-third solution (5% detergent + 10% alcohol + H_2_O). The traps were embedded in the ground, and the rim of the cup was flush with the ground. Arthropods in the traps were collected the next day and taken to the laboratory for identification and counting.

### 4.4. Statistical Analyses

Five indices were used to analyze the diversity of the arthropods: the species richness (*S*), the Shannon–Wiener diversity index (*H*), the Pielou evenness index (*J*), the dominance index (*C*) and the community similarity index (*C′*) [25,26,28,53].

The *S* is the number of species within a defined region.

The *H* of the arthropods was calculated using Equation.
$$H=-\overset{S}{\underset{i=1}{\sum }}P_{i}lnP_{i}$$

The *J* of the arthropods was calculated as per equation.
$$J=\frac{H}{lnS}$$

The *C* of the arthropods was calculated as per equation.
$$C=\overset{n}{\underset{i=1}{\sum }}{\left(\frac{N_{i}}{N}\right)}^{2}$$

The *C′* of the arthropods was calculated as per equation.
$$C^{\prime }=2w/\left(a+b\right)$$

*Pi* is the proportion of individuals belonging to a certain species, genus or family to the total number of individuals. *S* is the total number of species in a community. *Ni* is the number of individuals belonging to a certain species, genus or family and *N* is the total number of individuals in the region. *w* is the number of species shared by two samples, and *a* and *b* are the total numbers of species in each of the respective samples. Since it was not possible to identify all species in the field, the number of families was analyzed.

The mean values and SE of each community were calculated and compared using Student’s *t*-test to detect significant differences between XY and NT, NT and T1, and NT and T2. A significance level of *p* < 0.05 was used.

## Figures and Tables

**Figure 1 plants-11-02254-f001:**
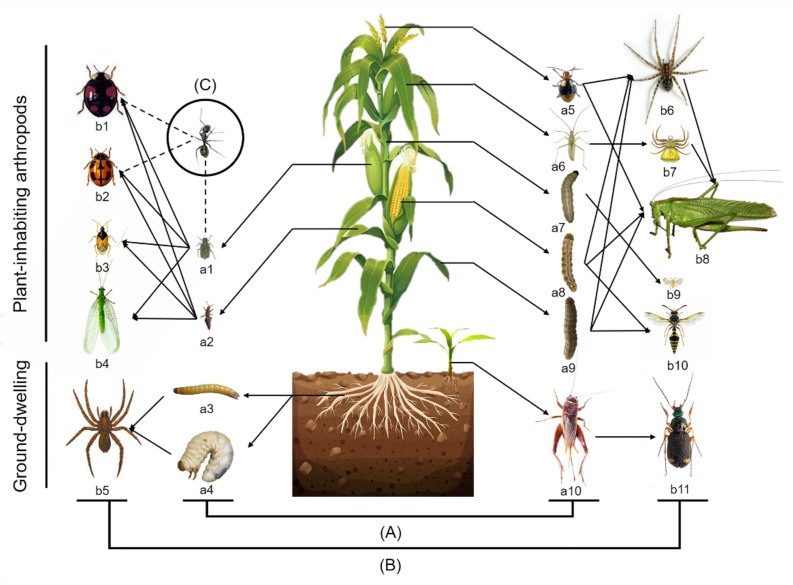
Important arthropod species in the food webs in maize field in Yitong, northeast China. (**A**) The herbivores included *Rhopalosiphum maidis* (**a1**), *Haplothrips aculeatus* (**a2**), *Monolepta hieroglyphica* (**a5**), *Trigonotylus ruficornis* (**a6**), *Ostrinia furnacalis* (**a7**), *Helicoverpa armigera* (**a8**), *Mythimna separata* (**a9**), eating leaves and buds, and wireworms (Coleoptera: Elateridae) (**a3**), *Anomala corpulenta* (**a4**), *Teleogryllus emma* (**a10**)*,* eating roots and stems; (**B**) the predators and parasites were *Harmonia axyridis* (**b1**), *Propylaea japonica* (**b2**), *Orius sauteri* (**b3**), *Chrysopa pallen* (**b4**)*, Pirata subpiraticus* (**b5**)*,* funnel-web spiders (Araneida: Agelenidae) (**b6**), *Misumenops tricupiuatus* (**b7**), *Tettigonia chinensis* (**b8**), *Trichogramma ostriniae* (**b9**), potter wasps (Hymenoptera: Vespidae) (**b10**) and *Chlaenius bioculatus* (**b11**); (**C**) another related arthropod was *Camponotus japionicus*. The solid arrows indicate predation relationships; the dashed arrows indicate regulatory relationships.

**Figure 2 plants-11-02254-f002:**
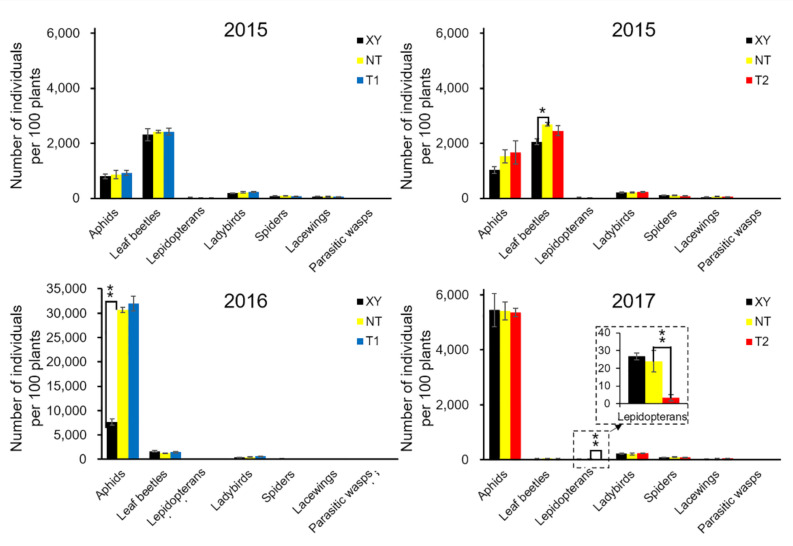
From 2015 to 2017, the cumulative number of main arthropod groups in the field of cultivated variety XianYu335 (XY), non-transgenic recipient variety DBN318 (NT), transgenic maize DBN9868 (T1) and DBN9936 (T2) over the growth season. The Y-axis shows the cumulative number of individuals per 100 plants during the growth period, and the X-axis shows the arthropod groups. We only compared XY with NT and NT with T1 or T2 by using a Student’s *t*-test. Asterisks denote significant differences, where *p* < 0.05 (*), *p* < 0.01 (**).

**Figure 3 plants-11-02254-f003:**
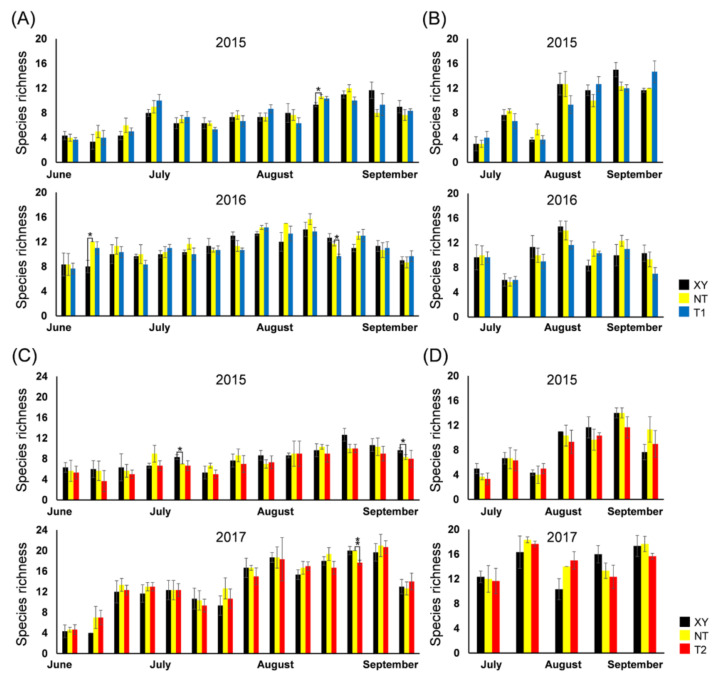
From 2015 to 2017, the species richness (*S*) dynamics of the arthropod community in the plots of cultivated variety XianYu335 (XY), non-transgenic recipient variety DBN318 (NT), transgenic maize DBN9868 (T1) and DBN9936 (T2). (**A**) The *S* of plant-inhabiting arthropods in XY, NT and T1; (**B**) the *S* of ground-dwelling arthropods in XY, NT and T1; (**C**) the *S* of plant-inhabiting arthropods in XY, NT and T2; (**D**) the *S* of ground-dwelling arthropods in XY, NT and T2. Y-axis: Mean ± SE (*n* = 3) of *S* at each sampling point. We only compared XY with NT and NT with T1 or T2 by using a Student’s *t*-test. Asterisks denote significant differences, where *p* < 0.05 (*), *p* < 0.01 (**).

**Figure 4 plants-11-02254-f004:**
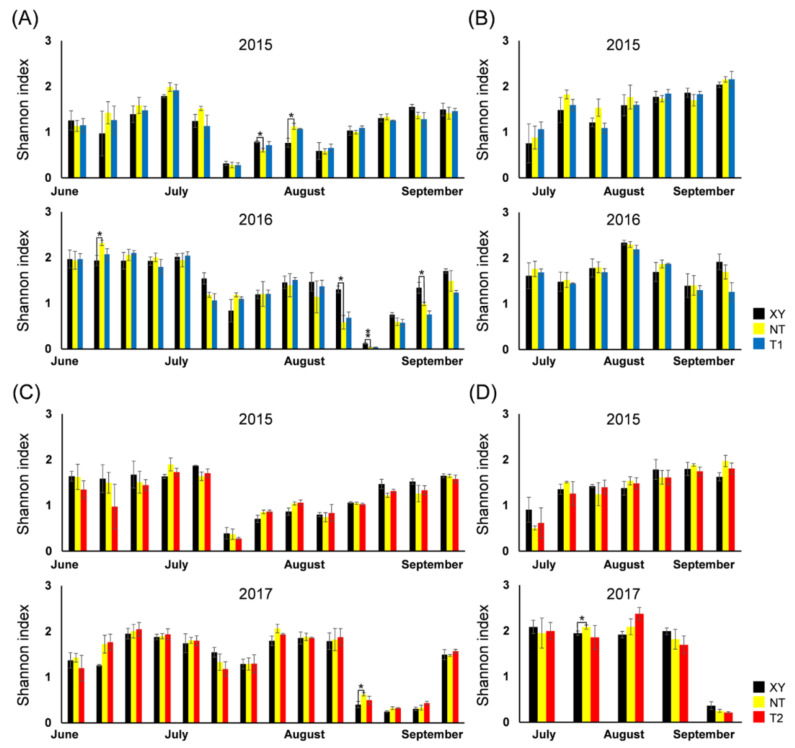
From 2015 to 2017, dynamics of Shannon–Wiener biodiversity index (*H*) of the arthropod community in the plots of cultivated variety XianYu335 (XY), non-transgenic recipient variety DBN318 (NT), transgenic maize DBN9868 (T1) and DBN9936 (T2). (**A**) The *H* of plant-inhabiting arthropods in XY, NT and T1; (**B**) the *H* of ground-dwelling arthropods in XY, NT and T1; (**C**) the *H* of plant-inhabiting arthropods in XY, NT and T2; (**D**) the *H* of ground-dwelling arthropods in XY, NT and T2. Y-axis: Mean ± SE (*n* = 3) of *H* at each sampling point. We only compared XY with NT and NT with T1 or T2 by using a Student’s *t*-test. Asterisks denote significant differences, where *p* < 0.05 (*), *p* < 0.01 (**).

**Figure 5 plants-11-02254-f005:**
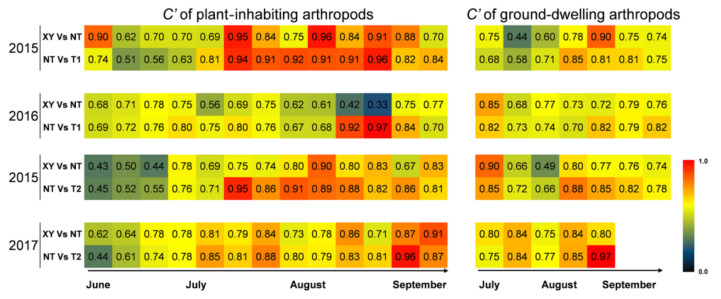
Heat maps of the community similarity index (*C′*) of arthropods in the plots of cultivated variety XianYu335 (XY), non-transgenic recipient variety DBN318 (NT), transgenic maize DBN9868 (T1) and DBN9936 (T2) from 2015 to 2017. Blue indicates a low value (0.0) of the community similarity index and red indicates a high value (1.0).

## Data Availability

Data is contained within the Appendix A. The data presented in this study are available in [Appendix A].

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
