# Peer review of "Transgenic Maize Has Insignificant Effects on the Diversity of Arthropods: A 3-Year Study"

_plants, 2022, doi:10.3390/plants11172254_

Round 1

Reviewer 1 Report

The authors provide data from altogether 4 field experiments, each with 3 different maize lines. Each experiment included one commercial non-related variety, the non-transformed parental line and a GM line, either expressing a herbicide tolerance trait (T1) or an insect resistance and HT trait (T2). Two sampling methods were used and more than 60 arthropod families were identified. The dataset is substantial and for sure worth of publication. However, the most important thing is the availability of the full data set so that they can be used for meta-analyses. There is currently no information where the data are deposited.

The results are focused on different diversity indices and the authors provide similar graphics of each index. This looks quite repetitive and takes a lot of space in the manuscript. If there is a way to combine those graphics or to move some graphics to the supplementary material, the paper would be easier to read.

A major point that needs considerable attention is the general English language, which is by far not adequate to native English. The paper needs professional language editing before it can be considered for publication.

In the following, I raise several individual points, but I did not correct English spelling and grammar.

ll. 31-32: I don’t understand the first sentence

l. 38: better: One concern about GM crops

l. 41: what is the difference between biological control and arthropod population regulation? And what is the difference between decomposition and recycling of organic matter?

l. 49: I would like to draw your attention to a very recent systematic review on Bt maize and non-targets: https://doi.org/10.1186/s13750-022-00272-0

ll. 50-56: Many Bt maize field studies lasted 3 years. I wonder why those two studies (17 and 18) have been cited and not others? Seems random to me.

ll. 57-65: Please specify for which kind of data the substantial equivalence approach is used. I think it is primarily for food and feed safety tests, not for environmental data. This should be made clear.

l. 84: did you determine all arthropods to family level? Or to species level (see l. 301)? Is it even possible to determine all maize arthropods to species? If species level data are available, a full species list in the supplementary material would be helpful.

l. 89: Borer

ll. 88-94: here you report on species. Did you also determine arthropods to species level? Please make clear in the Methods to which level you determined the taxa.

Figure 1: Where do the relationships between herbivores and natural enemies (arrows between taxa) come from? Is there any literature to cite? Please make this clear in the Methods section.

Figure 1: Make sure the species names are in italics

Figure 2: Are the values cumulative over the growth season? (Date 1 + Date 2 + Date 3 1 …)? Please specify.

Figure 2: Were there two separate experiments in 2015 (Experiment 1: XY, NT, T1; Experiment 2: XY, NT, T2) with the treatments XY and NT in each of the experiments? If yes, please make this clear in the Methods (ll. 288-292). Also, could you specify if the two experiments in 2015 were next to each other in different fields and at which distance?

Figure 2: Why is the bar for Lepidoptera in 2017 white?

Figure 2: It should be Lepidoptera or Lepidopterans. It should also be parasitic wasps and not sarasitic

l. 138/164/179/192: Asterisk

l. 152 what is communication structure information?

l. 215: in science it is difficult to prove something. I suggest to reword: there was no evidence that suggests that…

l. 222: I have never heard about bracts in the context of maize… what do you mean here?

ll. 240-256: The authors have a longer discussion on climatic effects on arthropods. However, they did not record any climatic parameters, so the discussion here is very speculative. Maybe this paragraph can be integrated into the previous one and shortened to one or two statements that climate is likely to have higher impact than the presence of the transgenes.

l. 260: what are intuitive evaluation results? I don’t understand the meaning.

l. 271: what do you mean by “the food chain of plant-aphid-ladybird is relatively independent within the food web”? I think aphids are a basic food for many natural enemies, so I don’t understand this.

Discussion in general: The authors should discuss the difference (or similarity) between effects of T1 and T2. T1 has no insecticidal gene, T2 has Cry1Ab. So if the Bt protein is toxic to nontargets, we would expect those effects in T2, but not in T1. For herbicide tolerant maize, no effects on nontargets are hypothesized. As there was not much difference in effects of both lines, it shows that Bt effects are unlikely. I think this difference between T1 and T2 should be discussed more prominently.

ll. 280-286: Please characterize the used maize lines further: are the GM lines planned to be commercialized, or are those lines experimental? What is the name of the transformation event? Are the used lines among the 3 events with certificates from the Chinese government? It think it is very important to provide information on the expression levels of Cry1Ab in T2 (and also on the herbicide tolerance traits) to demonstrate that the plants worked and that there was exposure of the nontargets to the transgenes. This might be ELISA data, PCR data, or a reference to other papers that measured expression of the used lines. For commercial lines, it can be assumed that expression was appropriate, but for experimental or non-commercial lines some sort of evidence is needed.

L. 307: how much alcohol, how much detergent?

Supplementary Table S1: I suggest to move the dates of the ground-dwelling collections further down so that the same dates are on the same line with the plant inhabiting arthropods.

l. 331: Did you only compare XY with NT and NT with T1 or T2? The comparison XY with T1 and T2 was not done? If there is a reason why this was done, I suggest to make it also clear in the results, otherwise the reader wonders why certain differences between XY and T1 or T2  in the figures are not significant…

Table S2: I think a full species list would be better than a list on order and family level.

Table S3: How did you define “main arthropod groups”? Please specify

Availability of raw data: The authors mention that their studies will provide baseline data for further ecological risk assessments of transgenic crops. The dataset is very comprehensive and valuable in particular for meta-analyses, such as this: https://doi.org/10.1186/s13750-022-00272-0. However, to be useful for meta-analyses, it is necessary to have access to the full dataset of the study. Most scientific journals, including MDPI journals ask for the publication of full datasets. You may either upload the data to a repository, then it is even citable, or you may simply attach a csv or excel file as a second supplementary material.

Reviewer 2 Report

The authors report of the biodiversity of GM crops in China from 2015-2017 and conclude no negative impacts from the GM crops.  This is a relatively well written manuscript.  Several comments:

Abstract:  

Line 25:  please choose a better word than "hardly".  This reviewer would suggest either "significant" at (P<0.05) or "not significant" (P>0.05)

This reviewer wonders why it took five years from the conclusion of these experiments (2015-2017) to submit this manuscript.  How many years has GMO maize with these traits been commercialized in this region in China?Can the authors provide evidence that their conclusions are still valid in 2022?

Reviewer 3 Report

The manuscript #plants-1812178 entitled ”Transgenic maize has insignificant effects on the diversity of arthropods: a 3-year study “ represent a long-term study of the influence of GM maize expressing Bt toxin gene and genes for herbicide tolerance, on the diversity of arthropods belonging to different orders and families. Taking into account important roles of arthropods in natural and farmland ecosystems such studies are important for evaluation of environmental safety of GM plants. The manuscript needs revision before acceptance for publication. Please find below my comments and suggestions.

ABSTRACT

-Line 14: Explain that PAT and EPSP genes provide tolerance to herbicides and Cry1Ab gene tolerance to lepidopteran insect pests.

-Line 16: Is it 17 and 67 as you mention in line 84?

KEYWORDS: Insert “maize” after “modified”.

INTRODUCTION

-Line 35: Space is missing after “area”. Please check spaces in the whole text, especially in References.

-Lines 54, 63: Do not start sentence with “And”. Replace with “Also,” or some other word.

-Line 60: Replace “considers” with “consider”.

-Lines 71-72: I think that you should mention tolerance of each transgenic maize

RESULTS

-Line 90: Replace “Boer” with “borer”.

-Caption to Figure 1: Latin names of species should be in italic. It is not clear which species are in the picture. For example, where is Rhopalosiphum maidis in the picture? Maybe you can mark them with numbers in the picture and then put that numbers in the caption to Figure at the species names. Also, meaning of crossed lines as well as meaning of full and dashed lines are not explained.

-Line 106: Do you mean “The number of species…….was lower….”?

-Line 109: It should be “bioculatus”.

-Figure 2: What is white bar in the sub-figure for 2017?

-Caption to Figure 2: What is culTable 335? Explain probabilities – Which test? Which comparison? Apply this in other figures.

-Lines 122, 141: t-test is not mentioned in Material and Methods.

-Line 126: Is it Lepidopteran?

-Figure 3: It is not clear which bars refer to june, which to july etc. Why some error bars are up and some are up and down (I sow this in other figures).

-Caption to Figure 3: It should be “asterisk”. Correct this in other captions.

-Caption to Figure 4: Replace “Dynamic” with “dynamics”.

-Line 168: Total number of individuals or…?

-Caption to Figure 5 and Figure 6: What is “Figure 335”?

-Table S3: Correct “Sarasitic” and “Lepidopteron”. Put table in landscape orientation.

DISCUSSION

-Lines 232-234: It seems that sentence is not complete.

-Line 237: “showed” instead of “shown”.

-Line 237-239: Can you provide some explanation/suggestion for this result.

-Line 240: Do you have data on climate factors in different years? You can put such data in supplement.

-Line 250-253: Please cite reference for this investigation.

-Line 65: Insert “was” after which”.

-Line 276: “to plant” instead of “to planting”.

MATERIAL AND METHODS

-Line 315: Give formula/description for S before H.

-Lines 316 and 318: Put “ln” instead of “Ln”.

-Line 326: Total number of what?

-Lines 326-329: a,b,w are not well explained. What are communities A and B? What are quantitative measures? What is “similar sum of…..”?

-Line 330: Is it SD or SE? In captions to Figures it is SE. Is it “The mean values and SE for diversity indices in each community……”?

-Line 331: What did you compare with ANOVA? Is it XY, NT and T1(or T2) within each sampling date? You should explain this. After ANOVA you should apply some post hoc test. In Results section you mention t-test. Please, explain this.

Round 2

Reviewer 1 Report

The authors revised the manuscript thoroughly and my main points in the previous version were addressed. 1) Raw data are supplied in supplemental material – thank’s a lot for that, I feel this is really important; 2) English language has been edited. However, I still found  a few words that I think should be changed (addressed below).

One issue remains for fixing and that is the discussion. I feel that several chapters in the discussion cover the same content. The authors should clearly separate the content in the different chapters (e.g. don’t discuss differences between GM and non-GM in a chapter on variety differences), or alternatively, don’t divide the discussion in those sections… see details below.

Comments refer to the pdf with mark-ups.

l. 23-24: the differences between ground-dwelling arthropod communitied were less obvious than those between plant-inhabiting communities

l. 28: had no significant effect

l. 39: by the Ministry

l. 55: impact of Bt maize producing Cry1Ab on

l. 58: that Bt maize (Cry1Ac) OR that Bt maize producing Cry1Ac

l. 60: that Bt maize

l. 65: concepts arising from the (delete coming)

ll. 78-82: I suggest to make two sentences here: In this study, the arthropod biodiversity in plots with four kinds of maize was investigated in Northeast China from 2015 to 2017. The maize cultivars were commercial maize XianYu335, non-transgenic maize DBN318, transgenic maize DBN9868 providing tolerance to herbicides, and transgenic maize DBN9936 providing tolerance to herbicides and resistance to lepidopteran pests.

L. 93; species in maize plots were

l. 96: replace “brief” with “simplified”

l.102: minute pirate bugs

l. 105: delete “in maize field”

l.117: Another related arthropod was

l. 125: summed data

Figure 2: please check that the y-axis text is not cut by the numbers (on my computer the text “per 100 plants” is cut at the bottom. Thank’s for clarifying the white bar representing Lepidoptera. Please make sure to use uniform labelling “Lepidopterans” instead of Lepidopteron, also in the dashed box.

l. 138: during the growth period

l. 142: in the plots of

l. 147: biodiversity indices of arthropods

l. 148: indices… The species richness (S) directly represents the…

l. 150: index (H) reflects

l. 155: abundance

l. 158: in the plots between

l. 172: plots of

l. 180: on different sampling dates

l. 186: in the plots of

l. 187: The H of plant- (capital T)

l. 191: at each sampling point.

l. 209: Figure 5

ll. 210 / l. 213: Here you mention the comparison XY vs. T1 and T2, in the previous figures you argue that it makes more sense to compare XY with NT and NT with T1 and T2. Why is the comparison XY with T of interest here, but not for abundance and biodiversity indices? Or is this simply a typo and should be NT vs. T?

l.218 plots

l. 230 plots

l. 232 husks of the ears

l. 236 outbreaks occur more frequently

l. 240-242: the few significant differences in arthropod biodiversity between GM and non-GM maize plots do not indicate that GM crops have an adverse effect on the ecosystem.

ll. 223ff. It is not clear in the discussion what the difference between the sections is. It seem that the text is repetitive here. I suggest to move the text from l. 225 to l. 228 before the subheading 3.1. This text is something like a summary of the results and the main conclusion. Then, the authors should focus on the differences between XY and NT and discuss the differences between NT and T in the following section.

L. 229-230: the differences between arthropod communities between the different maize lines were not caused

ll. 236.242: Once more, differences between transgenic and conventional is discussed here. Move to next paragraph?

l. 250: have shown the presence of Cry toxins

ll. 250-259: Much discussion about ground-dwelling arthropods. Should there be equal discussion of plant-dwelling arthropods? Toxin is also present in those species except aphids, which do not consume Cry proteins via phloem sap.

l. 313: Agriculture

l. 315: you may include the plot design in the supplemental material. It will help the reader to understand the two different experiments planted next to each other but including partly the same cultivars.

L. 333: the food consumption behaviors

l. 337 and throughout the manuscript: I suggest to use the proper symbol for diameter: Ø

Supplemental material: please make sure that tables are nicely fitting on pages (not one line or heading on other page etc.)

Table S4: did you determine expression levels in this study? If yes, I think some more information on methods (number of samples, sampling dates, used ELISA kits, etc.) would be good, otherwise a reference to a previous study if those values come from another study.
